# Morphological characteristics convey social status signals in captive tree sparrows (*Passer montanus*)

**Ju-Hyun Lee, Ha-Cheol Sung**⬤*

Department of Biological Science, Chonnam National University, Gwangju, The Republic of Korea

* shcol2002@jnu.ac.kr

## Abstract

In social animals that form flocks, individuals compete or cooperate to gain access to shared resources. In particular, group-foraging individuals frequently engage in aggressive interactions with conspecifics, including threat displays and physical attacks, in order to acquire food resources. Here, we investigated social interactions in flocks of captive tree sparrows (*Passer montanus*) to observe the formation of dominance hierarchies. We also examined correlations between social status and morphological traits to identify which physical traits act as indicators of dominance. To do so, we recorded aggressive behaviours (attacks and threats) of tree sparrows caught in two distinct regions in the Republic of Korea (Gwangju and Gurye). After merging the two groups, we examined dominance structures using David's scores for one month, and we recorded 1,051 aggressive interactions at a feeder in a group of 19 individuals. Using the number of aggressions and attack and threat behaviours, we tested whether morphological traits and sex influenced dominance structures. Aggressions were significantly more frequent in males than in females. However, no significant difference was observed the frequency of between- and within-sex aggression. In addition, differences in the frequency of aggression behaviours were observed between capture-site groups. Dominance structure was significantly correlated with certain morphological traits; specifically, the frequency of attacking behaviours was correlated with bill-nose length, and the frequency of threat displays was correlated with sex and badge size. These results suggest that social signals are closely related to morphological traits that are used to form dominance hierarchies in tree sparrow flocks.

## Introduction

In groups of social animals, individuals compete or cooperate to gain access to shared resources such as mating partners, food resources, and nesting sites, which affect individuals' fitness [1–3]. The benefits of flocking include better exploitation of food resources as well as detection and mobbing of potential predators. However, flocking entails costs such as intraspecific competition for limited resources and potentially harmful conflicts between individuals [4, 5]. Aggressive behaviour can incur costs when physical fights lead to bodily harm [6]. To

**Data Availability Statement:** All relevant data are within the paper.

**Funding:** This study was supported by a project entitled "Global Ph.D. Fellowship (2019H1A2A1073960)" The funders did not play

any role in the study design, data collection and analysis, decision to publish, or preparation of the manuscript.

**Competing interests:** The authors have declared that no competing interests exist.

minimize such costs, social species frequently resolve conflicts through alternative mechanisms through establishing dominance hierarchies [7] or through social status signalling [8]. These mechanisms help establish the order of access to resources with respect to an individual's dominance status and may adjust an individual's behaviour in a way that allows them to avoid unnecessary conflicts in social flocks [9].

Dominance hierarchies in groups emerge through dyadic aggressive interactions, producing a dominant individual (winner) and a subordinate individual (loser) [10]. Dominance hierarchies can regulate individual precedence and control aggressive behaviours among individuals with respect to their access to resources [4, 11]. Dominance hierarchies occur in various social species of birds in both captive and field conditions. Examples include white-throated sparrows (*Zonotrichia albicollis*; [12]), sociable weavers (*Philetairus socius*; [9]), house sparrows (*Passer domesticus*; [13]), and tree sparrows (*Passer montanus*; [14]). These species typically form foraging flocks, and individuals frequently compete for food resources through aggressive interactions.

In social flocks, most agonistic interactions are asymmetric due to opponents' unequal fighting abilities [15, 16]; and flock members may convey their social status through various signals. Generally, factors affecting dominance in birds include sex [4, 17], body size [18, 19], age [20], and plumage ornamentation [9, 21, 22]. Some morphological traits have been considered hierarchy signals with respect to physical fitness which includes male reproductive fitness, health, fighting ability, and aggressiveness [15]. Thus, each individual in a flock could maximize their physical fitness by predicting social status among flock members, leading to the evolution of status signalling that provides phenotypic correlates of dominance in a social flock [9].

The social status signalling hypothesis predicts that a dominance hierarchy is predetermined by adaptive variation in individual appearance [4, 23]. Social status signalling helps reduce unnecessary aggressive encounters and, in flocking bird species, plumage variability has evolved to convey hierarchy information [1]. A higher dominance rank may be exhibited by larger or more colourful patches [22], and melanin-based plumage coloration, (also referred to as 'badge of status') [9]. Status signalling has evolved to provide an 'honest signal' of competitive ability in social species that frequently compete in flocks. In the Passeridae and Emberizidae families, birds often exhibit black chest and throat patches that function as badges of social status [1, 22, 24]. In house sparrows, badge size is a reliable indicator of dominance in males and females, and large-badged individuals frequently attack small-badged individuals, but rarely vice versa [13]. In great tits (*Parus major*), breast stripe width is closely correlated with social status, and individuals with wider stripes are more aggressive [25]. Eurasian siskins (*Carduelis spinus*) may bear a badge of dominance with individuals with larger, blackish bib plumage are typically dominant over individuals exhibiting smaller badges. These large-badged individuals were found to win 77% of agonistic interactions [26]. The study of sociable weaver under the field condition, they exhibit an ordered hierarchy within colony and the size of the bib predicts success in social competition In *Passer* species [9], the size of the melanin-based black throat patch (i.e., badge) is the most reliable predictor of dominance rank (i.e., a status signalling trait), individuals with large badges are more likely to win aggressive encounters than individuals with smaller badges [13, 14, 27–29].

In social birds, conflict between groups (intergroup conflict) occurs for the occupation of several resources [30, 31]. Especially, cooperatively breeding birds often compete against individuals from other groups in order to protect their territories. However, such competition can differ between breeding and non-breeding season [32]. When intergroup conflicts were observed, group size and home field advantage influenced the outcome of competition [30, 31]. However, the role played by group selection in such circumstances in unclear.

The tree sparrow is a social sparrow and is widely distributed in the Republic of Korea [33]. These birds form social foraging flocks from the end of the breeding season through winter, and flock members typically forage and roost together during winter [34, 35]. Individuals in flocks frequently exhibit intraspecific aggressive behaviour to gain access to resources (e.g., food resources, foraging sites, and roosting sites) and they establish dominance hierarchies within their flocks [14, 36]. A previous study observed linear dominance hierarchies in two of three captive tree sparrow flocks, but the status signalling function of badges was not clearly supported, as badge size was a reliable predictor of dominance rank in only one flock [14]. However, a different study suggested that throat patch size acted as a social status signal in males but not females, as fighting success and badge size were correlated only in males [29]. The various competitive interactions between individual tree sparrows have not been examined by distinguishing between aggressive and threat behaviours. We thus examined dominance hierarchies based on aggressive behaviours in captive tree sparrow and do distinguish between attack and threat behaviours to test the occurrence of social status signalling.

## Materials and methods

### Study site and captive conditions

This study was conducted using captive tree sparrows caught at two sites, located 60 km apart. Tree sparrows were captured using a mist net (nylon 30 mm mesh; Avinet, Portland, ME, USA) in March 2020 at Gwangju (GJ; 879, Pochung-ro, Nam-gu, Gwangju, Republic of Korea; 35˚5'15.94" N, 126˚51'36.35" E) and Gurye (GR; 261–1, Hwangjeon-ri, Masan-myeon, Gurye-gun, 35˚14'5.46" N, 127˚29'22.84" E). After capturing tree sparrows, we measured the following morphological variables according to *the Bird Banding Manual* [37]. Tarsus and beak lengths were recorded to the nearest 0.1 mm using a BLUETEC digital calliper, and wing length (wing chord length) and body lengths were measured to the nearest 1.0 mm using a ruler. Beak lengths include bill-nose length (length from the tip of the nostril to the tip of the beak), bill-head length (length from the tip of the back of the head to the tip of the beak), and bill-skull length (length from the junction of the skull and beak to the tip of the beak). Body length is the length from the tip of the beak to the tip of the tail when the bird is laid down, and badge size was defined as the area covered by black feathers on the throat. We photographed each individuals using a digital camera (Samsung Galaxy S20+, Samsung, Suwon-si, Republic of Korea), and measured badge size (mm$^2$) using ImageJ software (version 1.52 for Windows; [38]). Each individual was photographed from a distance of 15 cm with its back facing a black wall. To facilitate quick recognition of individual tree sparrows during the behaviour recordings, we applied different combinations of coloured bands (Avinet; Darvic markers XF 2.3–2.5 mm) to both legs of each individual and applied corresponding colour markings to the tail feathers. Birds were sexed by PCR-based amplification of the sex-chromosome-linked *CHD1* gene using p2 and p8 primers [39].

We transferred captured tree sparrows to a semi-outdoor aviary on the rooftop of Natural Science Building #1, Chonnam National University, Republic of Korea (35˚10' 38.06" N, 126˚54'33.50" E). The size of the aviary was 3.5 m × 6 m × 2 m, and we installed four cameras to record the sparrows' behaviour during the daytime. The aviary contained seven nest boxes, 1.5-m high roosting trees, a food dish, a water dish, and a sand dish, and it was illuminated by natural light. Water and food were provided *ad libitum*. We placed a round feeding plate (a white plastic plate, 50 cm in diameter) containing food (i.e., a mixture of millet seeds, vitamin-enriched cracked corn, and mealworms) at a fixed location in the centre of the aviary. Nineteen tree sparrows (six males and four females from GR and three males and six females from GJ) were housed in the aviary. We released captured individuals into the aviary after stabilizing

in a dark room, and behavioural recording was started one week after the captive individuals had adapted to the aviary without human access for a week. Behavioural recordings were conducted during the daytime (06:00–19:00) from March to May before the breeding season, and the recording was stopped at the start of the breeding behaviour.

## Behavioural analyses

We recorded the total number of aggressive behaviours of tree sparrows that occurred on the feeding plate and in the surrounding area. Aggressive behaviours were classified into two categories: attacks and threat displays. Attacks were defined as physical contact involving pecking the body, kicking, and holding the opponent using the bill. Threat displays were defined as non-physical contact used to displace an opponent (e.g. distracting movements, fluffing feathers, head-forward threat display, etc.). We recorded aggressive behaviours, with respect to attacks and threats, and we identified 'the winner' and 'the loser' of each interaction if one individual unambiguously displaced the other. We omitted and ignored ambiguous displacements. If attacks occur simultaneously between each other in the X-to-Y dyad, both X and Y are considered winners and both scored 1. 'Fighting success' was recorded as a binomial variable comprising the total number of aggressive interactions; thus, it was possible to record winning and losing situations between individuals to generate interaction matrices based on paired comparisons.

Dominance relationships were determined by the outcomes of aggressive behaviours within each dyad. To determine the dominance rank of individuals within the flock, we used David's score (DS; [40]) to calculate dominance scores for the 19 individuals. Individual DS ranks have the advantage that win/loss asymmetries are considered by integrating the dyadic dominance proportions in the calculation; therefore, it is not disproportionately weighted by minor deviations from the main dominance direction [41] DS was calculated by $P_{ij}$ (the proportion of wins by individual $i$ in its interactions with individual $j$) and $P_{ji}$ (the proportion of wins by individual $j$ in its interactions with individual $i$) of an individual weighted by the relative strength of its opponents. The $P_{ij}$ is the number of times that $i$ defeats $j$ ($a_{ij}$) divided by the total number of its interactions between $i$ and $j$ ($n_{ij}$; $P_{ij} = a_{ij}/n_{ij}$). Thus, DS for each member could calculated with this formula:

$$\mathbf{DS} = w + w_2 - l - l_2$$

where $w$ is the sum of the $P_{ij}$ values of $i$, $w_2$ is the summed $w$ values of those individuals that $i$ interacted with, $l$ represents the sum of $P_{ji}$ values of $i$', and $l_2$ represents the summed $l$ values of those individuals that $i$ interacted with [40, 41]. We calculated the DS values (DSs) in three forms: the number of aggressive behaviours ($DS_{aggression}$), the number of attacks ($DS_{attack}$), and the number of threat displays ($DS_{threat}$). Individual ranks were determined according to $DS_{aggression}$.

## Statistical analyses

We investigated whether the number of aggressive behaviours differed between sparrows of different sexes as well as different regional groups (GJ or GR). To compare frequencies of aggressive behaviours regarding sex and capture site, we used Mann-Whitney $U$ tests and Kruskal-Wallis tests. In addition, we also examined differences in the number of aggressive behaviours in two different tree sparrow demographic categories using a $t$-test; the first category was within-group or between-group, and the second category was same-sex or inter-sex. To investigate whether the DSs of a given individual were predicted by morphological traits, we fit a generalized linear model (GLM) using sex, wing length, tarsus length, bill-nose length,

and badge size. $DS_{aggression}$, $DS_{attack}$, and $DS_{threat}$ were the dependent variables, and sex was the fixed factor. In addition, wing length, tarsus length, bill-nose length, and badge size were used as covariates. Statistical analyses were conducted using the IBM SPSS statistics software package (v.21, IBM Corporation, Armonk, NY, USA). Statistical significance is reported at $\alpha = 0.05$. Results are presented as the mean ± standard deviation.

### Ethical note

This study was approved by the Chonnam National University (CNU) and CNU Laboratory Animal Research Centre (License no: CNU IACUC-YB-2020-01) under the Association for the Study of Animal Behaviour Guidelines for the Treatment of Animals in Behavioural Research. Tree sparrows received their typical daily diets, and water was available *ad libitum* throughout the observations. We made efforts to minimize disturbance during behavioural data collection by recording without the presence of researchers.

## Results

During the recording period, we observed 1,051 aggressive behaviours, including 503 attacks and 548 threat displays. Attack and threat displays were recorded the most in the first week, and attack displays rapidly decreased by week, but threat display decreased relatively slowly (Fig 1). The mean number of performed total aggressive behaviours per individual was 55.32 ± 47.54, and the mean number of attacks and threat displays was 26.47 ± 20.68 (1–70) and 28.84 ± 28.05 (0–86), respectively. The average number of aggressive behaviours per individual differed significantly between sexes, with males engaging in 91.63 ± 52.19 aggressive behaviours and females performing 28.90 ± 23.77 aggressive behaviours ($n = 19$; Mann-Whitney $U$-test: $U = 10.0$, $Z = -2.86$, $P < 0.05$). The proportion of dyads in aggressive encounters depended on the sex of the two opponents ($\chi^2_3 = 12.871$, $P < 0.05$). The number of aggressive encounters was the highest in male-to-female dyads, followed by male-to-male dyads and female-to-female dyads. The number of aggressive encounters in female-to-male dyads was the lowest (Table 1). No significant difference was observed between GR-to-GR dyads, GR-to-GJ dyads, GJ-to-GR dyads, and GJ-to-GJ dyads with respect to the number of aggressive behaviours ($\chi^2_3 = 0.073$, $P = 0.99$; Table 1). The number of aggressive behaviours between individuals from different capture-site groups was significantly higher when those individuals were of the same sex ($t = 2.31$, $P = 0.029$). However, this difference did not exist in within-group interactions between same-sex or inter-sex individuals (Table 2).

Higher-ranking individuals displayed significantly higher number of aggressions (Spearman's rank test; $r_s = 0.892$, $n = 19$, $P < 0.001$), attacks ($r_s = 0.889$, $n = 19$, $P < 0.001$), and threat displays ($r_s = 0.855$, $n = 19$, $P < 0.001$). High-ranking individuals (ranking from $1^{st}$ to $5^{th}$) performed more threat displays than attacking behaviours, whereas intermediate-ranked individuals ($6^{th}$ to $14^{th}$) showed more attacks than threat displays. Moreover, aggressions and threat behaviours were significantly correlated with sex (GLM estimate of $DS_{aggression} = 56.24 ± 21.67$, $t = 2.59$, $P = 0.022$; GLM estimate of $DS_{threat} = 73.91 ± 22.34$, $t = 3.31$, $P = 0.006$; Table 3, Fig 2), but not with the frequency of attacks (GLM estimate of $DS_{attack} = 31.46 ± 16.75$, $t = 1.88$, $P = 0.083$; Table 3, Fig 2). Regarding the correlation of dominance scores based on total aggressive behaviours and morphological traits, $DS_{aggression}$ was significantly correlated with tarsus length, bill-nose length, and badge size (tarsus length: $t = -2.16$, $P = 0.049$, B ± SE = $-28.17 ± 12.99$; bill-nose length: $t = 3.06$, $P = 0.009$, B ± SE = $33.90 ± 11.09$; badge size: $t = 3.06$, $P = 0.009$, B ± SE = $1.13 ± 0.42$), and males had higher $DS_{aggression}$ than females (sex: $t = 2.59$, $P = 0.022$, B ± SE = $56.24 ± 21.67$; Table 3, Fig 2A and 2B). However, $DS_{attack}$ and $DS_{threat}$ showed unique correlations with morphological traits. $DS_{attack}$ was

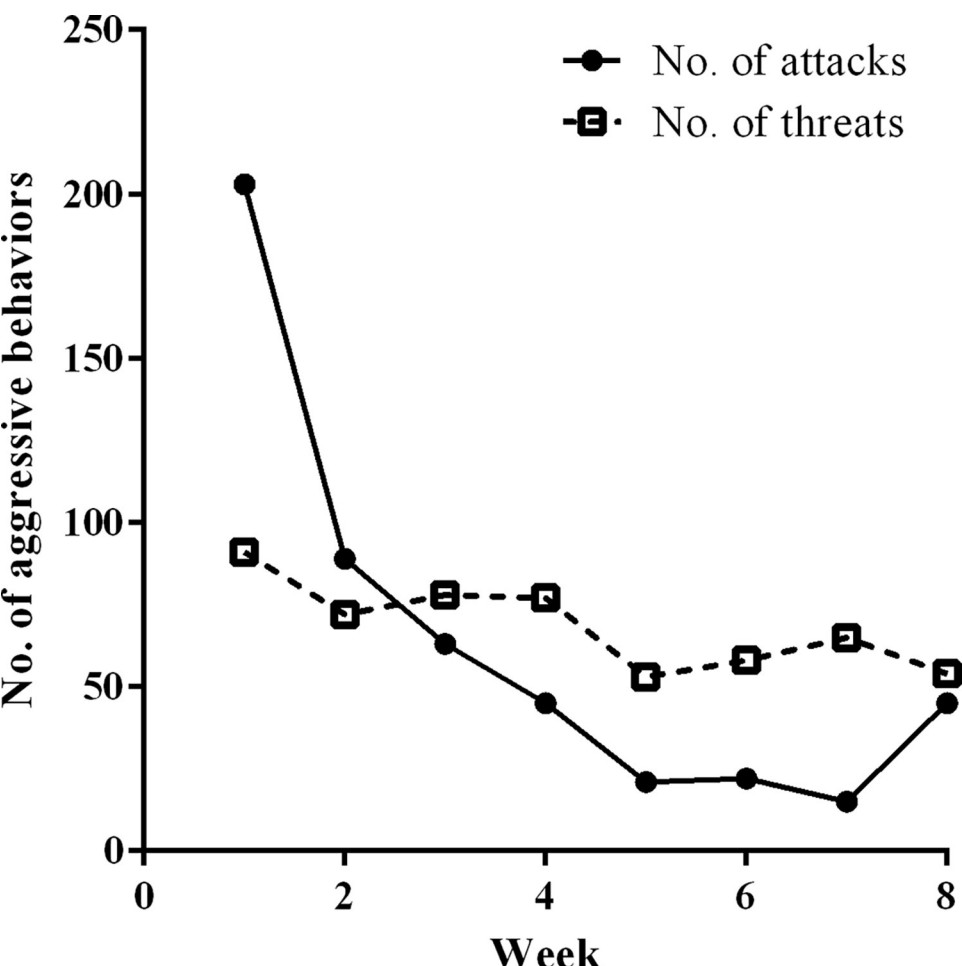

**Fig 1. Changes of the number of attack and threat displays by week.**

significantly correlated with bill-nose length ($t$ = 3.05, $P$ = 0.009, B ± SE = 26.10 ± 8.56), but not with other morphological traits or sex (Table 3, Fig 2C and 2D). DS$_{threat}$ was significantly correlated with badge size and sex, with males showing higher scores than females (badge size: $t$ = 2.27, $P$ = 0.041, B ± SE = 0.99 ± 0.44; sex: $t$ = 3.31, $P$ = 0.006, B ± SE = 73.91 ± 22.34), but it

**Table 1. Number of aggressive behaviours, categorized by sex-to-sex dyads and group-to-group dyads (referring to capture-site groups).**

|  | Category | Number of aggressions (mean ± SD) | $\chi^2$ | $P$ | $n$ |
|---|---|---|---|---|---|
| **Sex to Sex** | **M → M** | 41.50 ± 25.99 | 12.871 | **0.005** | 9 |
|  | **M → F** | 50.13 ± 28.70 |  |  | 9 |
|  | **F → M** | 9.64 ± 8.89 |  |  | 10 |
|  | **F → F** | 19.27 ± 15.84 |  |  | 10 |
| **Group to Group** | **GR → GR** | 33.10 ± 33.36 | 0.073 | 0.995 | 10 |
|  | **GR → GJ** | 29.40 ± 27.12 |  |  | 10 |
|  | **GJ → GR** | 23.33 ± 21.62 |  |  | 9 |
|  | **GJ → GJ** | 24.00 ± 15.52 |  |  | 9 |

SD: standard deviation, M: male, F: female, GJ: Gwangju, GR: Gurye.

**Table 2. Number of aggressive behaviours within and between capture-site groups.**

| Category 1 | Category 2 | Number of aggressions (mean ± SD) | t | P | n |
|---|---|---|---|---|---|
| Within Group | Within Sex | 16.05 ± 13.83 | -1.76 | 0.861 | 19 |
| | Between Sex | 16.89 ± 13.79 | | | 19 |
| Between Groups | Within Sex | 17.00 ± 12.57 | **2.31** | **0.029** | 19 |
| | Between Sex | 9.53 ± 6.42 | | | 19 |

was not significantly correlated with wing length, tarsus length, and bill-nose length (Table 3, Fig 2E and 2F).

## Discussion

Competition is essential for birds to occupy limited food resources, and agonistic interactions are common, especially in social birds living in groups. In this situation, the social system was characterized by dominance hierarchy to mediate conflicts and the social status signal has evolved to reduce the cost by agonistic interaction [42, 43]. When foraging on feeding plates, tree sparrows frequently interacted through aggressive behaviours, including attacks and threat displays and they established a dominance hierarchy based on the outcomes of aggressive behaviours. We found that morphological characteristics independently affected social status signalling, and it is likely to be important for organizing stable society to mediate conflicts in the tree sparrow flock.

Social species frequently establish dominance hierarchies to prevent conflicts during competition for limited resources [9, 29, 42, 43] and the observed agonistic interactions of tree

**Table 3. Effects of morphological traits on the outcomes of aggressions, attacks, and threat displays in tree sparrows.**

| | B ± SE | 95% CI | t | p |
|---|---|---|---|---|
| **DS$_{aggression}$ ($r^2 = 0.849$)** | | | | |
| Intercept | -113.15 ± 350.84 | -871.01–644.78 | -0.32 | 0.752 |
| Sex | 56.24 ± 21.67 | 9.42–103.07 | 2.59 | **0.022** |
| Wing length | 2.48 ± 3.96 | -6.08–11.04 | 0.63 | 0.542 |
| Tarsus length | -28.17 ± 12.99 | -56.25–0.09 | -2.16 | **0.049** |
| Bill-nose length | 33.90 ± 11.09 | 9.95–57.85 | 3.06 | **0.009** |
| Badge size | 1.13 ± 0.42 | 0.38–2.20 | 3.06 | **0.009** |
| **DS$_{attack}$ ($r^2 = 0.746$)** | | | | |
| Intercept | -155.06 ± 271.11 | -740.75–430.63 | -0.57 | 0.577 |
| Sex | 31.46 ± 16.75 | -4.72–67.64 | 1.88 | 0.083 |
| Wing length | 2.27 ± 3.06 | -4.35–8.89 | 0.74 | 0.472 |
| Tarsus length | -15.87 ± 10.04 | -37.56–5.83 | -1.58 | 0.138 |
| Bill-nose length | 26.10 ± 8.56 | 7.59–44.61 | 3.05 | **0.009** |
| Badge size | 0.46 ± 0.33 | -0.24–1.17 | 1.41 | 0.183 |
| **DS$_{threat}$ ($r^2 = 0.838$)** | | | | |
| Intercept | 59.68 ± 361.56 | -721.41–840.78 | 0.17 | 0.871 |
| Sex | 73.91 ± 22.34 | 25.65–122.16 | 3.31 | **0.006** |
| Wing length | -1.11 ± 4.08 | -9.93–7.71 | -0.27 | 0.790 |
| Tarsus length | -18.01 ± 13.39 | -46.95–10.92 | -1.34 | 0.202 |
| Bill-nose length | 23.88 ± 11.42 | -0.80–48.56 | 2.09 | 0.057 |
| Badge size | 0.99 ± 0.44 | 0.05–1.93 | 2.27 | **0.041** |

Results were obtained from a generalized linear model. B ± SE: coefficient ± standard error; CI: confidence interval.

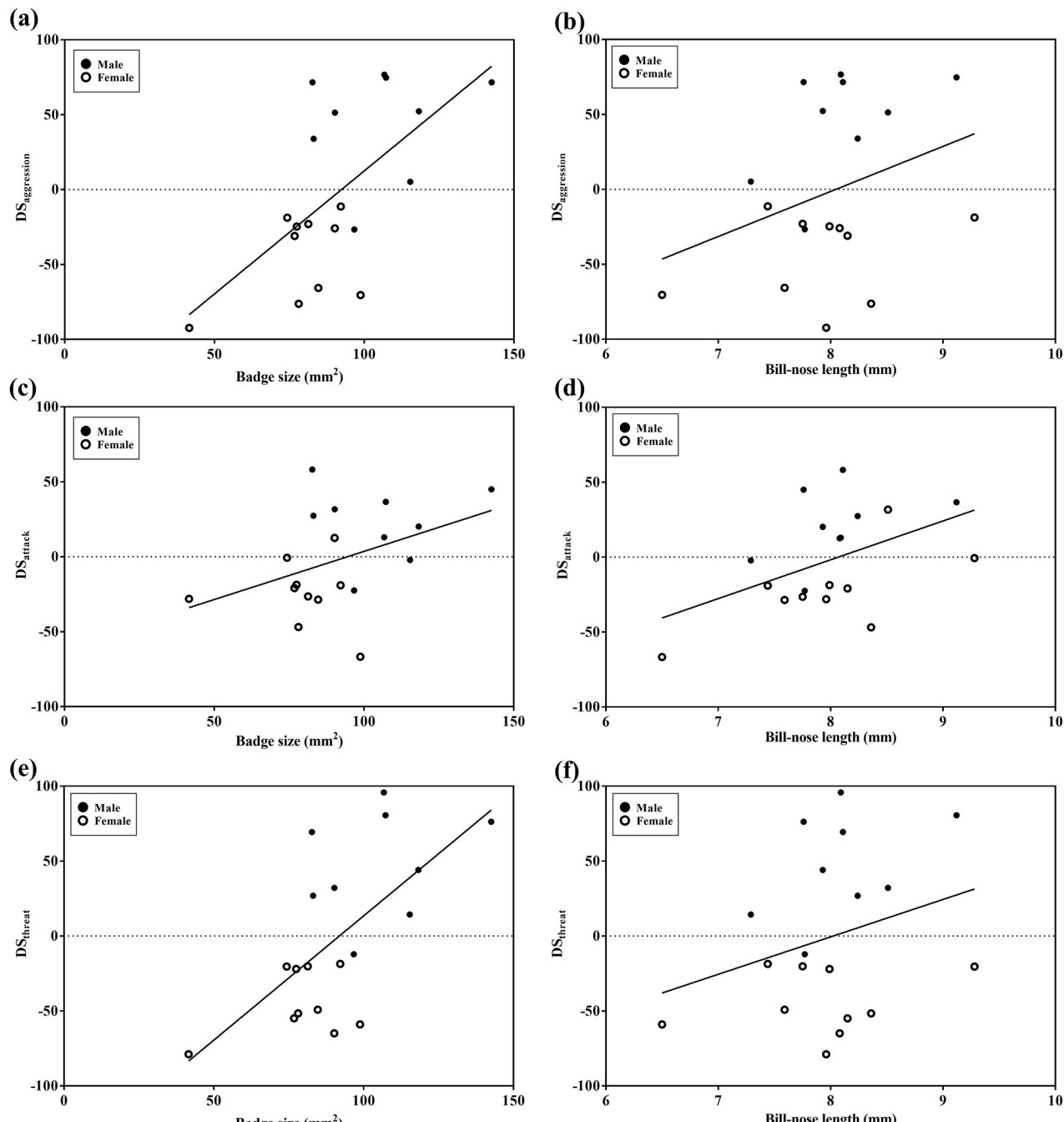

**Fig 2. The relationship between either morphological trait or DS scores.** The relationship between either morphological trait and badge size (a, c, e) or morphological traits and bill–head length (b, d, f) in relation to $DS_{aggression}$ (a, b), $DS_{attack}$ (c, d), and $DS_{threat}$ (e, f). Higher David's scores (DS) indicate more dominant individuals. Solid lines indicate the estimated effect of morphological traits on dominance scores.

sparrows were consistent with this prediction. Tree sparrows showed aggressive behaviours when competing over food resources in captivity, and dominance hierarchies were observed. During agonistic interactions, dominant individuals initiated aggressive behaviours more frequently than subordinate individuals. Such asymmetric dominance relationships and social organizations frequently occur in foraging flocks of tree sparrows [14, 29], house sparrows [13,

27, 44], and sociable weavers [9]. Considering that, in the present study, the number of aggressive behaviours of high-ranking individuals (mostly males) differed markedly from those of low-ranking individuals (mostly females), low-ranking individuals may reduce the costs of agonistic interactions involving competition for resources by avoiding unnecessary fights with more aggressive high-ranking individuals.

In many passerine birds, males initiate agonistic interactions and are dominant over females [9, 11, 29, 45, 46]. In the current study, males had higher dominance ranks than females, and males also exhibited agonistic interactions more frequently than females. Specifically, the proportion of aggressive encounters appeared in the descending order of male-to-female dyads, male-to-male dyads, female-to-female dyads, and female-to-male dyads. In addition, female-to-female dyads showed a significantly higher frequency of aggressive encounters than female-to-male dyads. This difference may have evolved to reduce conflict costs faced by females due to intersexual aggression and to increase the dominance status of the flock [4, 47, 48]. Evening grosbeaks (*Hesperiphona vespertina*) in flocks are more likely to engage in agonistic encounters with same-sex conspecifics than with those of the opposite sex. Particularly, females are more likely to fight with other females but they generally avoid fighting with males [4]. Similarly, in the current study, female tree sparrows rarely attacked males but attacked other females more often in competition for food. Therefore, our results support the asymmetry of intersexual agonistic interactions.

With regard to capture site, the number of agonistic interactions did not differ significantly between intra-group and inter-group encounters. There are two possible outcomes of encounters with unacquainted individuals from a different group: aggression [49] or preference (when meeting a new individual of the opposite sex) [50]. Great tits are more aggressive when encountering new conspecifics than when exposed to known individuals [49]. However, in the current study, tree sparrows did not show significant differences in the number of agonistic interactions between inter- and intra-group confrontations during the research periods. Tree sparrows are highly sociable birds that frequently form foraging groups of various sizes, the compositions of which change frequently ([29, 35] J.H.L. pers. obs.). Thus, tree sparrows frequently encounter new individuals, which may explain why no difference in aggression was observed with respect to capture-site groups. However, inter-group aggression towards the opposite sex was significantly less frequent, which is assumed to be due to preference for novel individuals of the opposite sex. According to [50]), Japanese quail (*Coturnix coturnix*) prefer conspicuous and slightly novel partners that somewhat different from familiar ones. Thus, tree sparrows did not show more frequent aggression towards individuals from other groups because of their high sociality, and they also showed less aggression towards potential mating partners from the opposite capture-site group.

In the present study, low-ranking individuals avoided high-ranking individuals, even in the absence of a direct physical attack. High-ranking individuals displayed more threatening behaviours than attacking behaviours, whereas individuals of intermediate ranking showed more attack behaviours than threat displays. Regarding agonistic interactions of social birds, threat displays are typically preferred over physical aggression to reduce the risk of injury [51]. High-ranking individuals commonly show conflict avoidance, and both dominant and subordinate individuals avoid physical fights to reduce competition costs (i.e., conflict management strategy; [9, 52–54]. Threat displays convey information [55] such as motivation and fighting ability [56–58]. Therefore, individuals of many species presumably advertise their fitness through initial displays or ornaments to assess the fighting ability of their opponents and avoid physical conflicts [59–61]. In the present study, high-ranking tree sparrows frequently displayed threat behaviours to establish a stable dominance hierarchy.

Bill length was a significant predictor of $DS_{attack}$, and sex and badge size were significant predictors of $DS_{threat}$. Fighting ability in birds generally depends on physical fitness, as these traits typically reflect individual strength [62, 63]. In addition, badge size is correlated with fighting ability and predicts dominance over male and female conspecifics [29, 64, 65]. Our results suggested that sex and badge size were correlated with threat behaviour, whereas bill length was associated with fighting behaviour. A previous study on status signalling in tree sparrows suggested that male fighting success was predicted by badge size, whereas female fighting success was predicted by wing length [29]. In tree sparrows, badge size is larger in males than in females. However, bill-nose length does not differ consistently between sexes [39]. In addition, badge size and bill length showed a positive correlation in males, but not in females [66]. Thus, sex and badge size are signals of high-status during threat displays, whereas bill-nose length may be used to determine fighting ability.

The 'badge of status' may have evolved in social groups as an honest signal used to showcase an individual's fitness to unfamiliar individuals in order to compete for limited food resources. However, badges may be less functional in interactions with familiar individuals [22, 44, 49, 65]. In the current study, in which we used individuals from two capture-site groups, badge size may have been an indicator of the fitness of unfamiliar individuals. In different species (including great tits and house sparrows) occurring in foraging flocks, plumage ornamentation does not predict an individual's success when fighting a familiar individual, but rather indicates fighting ability to unfamiliar opponents from a different group [44, 49]. In addition, tree sparrows frequently form large groups (sometimes hundreds of individuals), whose members were changed constantly. Thus, they may not be able to recognize and memorize the fighting ability of each member of their flock (see [9]). Badge-of-status signalling may help individuals gather reliable information on the dominance status of other group members, and it also helps an individual decide whether to threaten, fight, or flee.

In conclusion, tree sparrows exhibited a dominance hierarchy within the captive flock, and morphological traits correlated with dominance hierarchies deduced from aggressive behaviours, implying the existence of social status signalling. Bill-nose length correlated with attack behaviour, and badge size was related to threat behaviour. Thus, social stability of tree sparrow flocks may benefit from social status signalling as a conflict-resolution strategy.

## Acknowledgments

We are grateful to Kang-Sik Kim, Ju-Eun Lee, Yu-Jeong Oh, and Seung-Jun Oh for the help provided for the behavioural analysis. We also acknowledge members of animal behavior & ecology lab to participate aviary setting and managing.

## Author Contributions

**Conceptualization:** Ju-Hyun Lee, Ha-Cheol Sung.

**Formal analysis:** Ju-Hyun Lee.

**Investigation:** Ju-Hyun Lee.

**Methodology:** Ju-Hyun Lee.

**Project administration:** Ju-Hyun Lee.

**Supervision:** Ha-Cheol Sung.

**Visualization:** Ju-Hyun Lee.

**Writing – original draft:** Ju-Hyun Lee.

**Writing – review & editing:** Ju-Hyun Lee, Ha-Cheol Sung.

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
