## [Decision Letter · Decision Letter 0]

20 Feb 2023

PONE-D-22-22925Morphological characteristics convey social status signals in captive tree sparrows (Passer montanus)PLOS ONE

Dear Dr. Sung,

Thank you for submitting your manuscript to PLOS ONE. After careful consideration, we feel that it has merit but does not fully meet PLOS ONE’s publication criteria as it currently stands. Therefore, we invite you to submit a revised version of the manuscript that addresses the points raised during the review process.

As one of the reviewers pointed out, your study was done during the early breeding season when these birds usually do not live in groups. You need to refer to this issue in the method section and elaborate on its possible implications in the discussion.

I fully agree with the first reviewer that a temporal axis should be included in the data analysis to test if and how familiarity with other individuals influences the results.  

As pointed out by the second reviewer, it is not clear if the stress of being captured and transferred to an aviary influences the number or type of aggressive behaviors. It would be best if you referred to this possibility in the discussion.

In addition, you do not refer to similar studies done under field conditions. I find it hard to believe that there are no such studies, but if this is the case, clearly say it.

Finally, make sure that your paper is well-edited (see comments by both reviewers).

We look forward to receiving your revised manuscript.

Kind regards,

Ofer Ovadia

Academic Editor

PLOS ONE

2.We note that you have stated that you will provide repository information for your data at acceptance. Should your manuscript be accepted for publication, we will hold it until you provide the relevant accession numbers or DOIs necessary to access your data. If you wish to make changes to your Data Availability statement, please describe these changes in your cover letter and we will update your Data Availability statement to reflect the information you provide.

Reviewers' comments:

Reviewer's Responses to Questions

**Comments to the Author**

1. Is the manuscript technically sound, and do the data support the conclusions?

Reviewer #1: Partly

Reviewer #2: Yes

2. Has the statistical analysis been performed appropriately and rigorously? 

Reviewer #1: I Don't Know

Reviewer #2: Yes

3. Have the authors made all data underlying the findings in their manuscript fully available?

Reviewer #1: Yes

Reviewer #2: Yes

4. Is the manuscript presented in an intelligible fashion and written in standard English?

Reviewer #1: Yes

Reviewer #2: Yes

5. Review Comments to the Author

Reviewer #1: Intro

Line 40: “flocks” implies birds, but the authors make a point of saying “social animals”. Either say social animals and groups or flocks and birds.

41: “individuals’” implies multiple individuals. But the authors say “the” individuals’, which is a bit confusing. Maybe omit the “the”

43: Here you say “food resources”, whereas in the previous sentence (and in the abstract), the authors refer to food as “forage”. Be consistant.

53: do you really want to put “loser” and the citation in the same set of parentheses?

56: “several”? Perhaps you should say that hierarchies have been demonstrated in various bird species. The way it is currently stated implies that these are the only species that hierarchies exist in.

56: You say “social species”. Is that necessary? Can a dominance hierarchy exist in a non-social species?

61: comma splice. Perhaps it would be better to have a semi-colon followed by “flock members….”

65: is it necessary to say “reproductive fitness”? I’m assuming you are distinguishing it from “physical fitness”?

66: here you use the term “fitness” without qualifier – is it necessary to in the previous line?

73-74: you say “certain morphological traits” and then “larger or more colourful patches”. Arent the latter examples of the former?

And aren’t all of them social status signals?

74: need a comma before social status signals

76: maybe say “birds often exhibit….”

Or “individuals often exhibit….”

82: maybe say “with” instead of “and”

And maybe “sporting” or “exhibiting” rather than “with” later in the sentence

88: “are commonly occurred”?? Huh?

90: reword “compete for protecting territories”. Maybe “often compete against individuals from other groups in order to protect their territories. However, such competition can differ between breeding…”

92: not “were occurred”. Maybe say “were observed”

92: use past tense (influenced)

93-94: A bit confusing. Maybe “However, the role played by group selection in such circumstances in unclear.”

95: Have you introduced this species already in the Introduction? If not, provide scientific name.

95: what is meant by “a typical social bird”? I suggest dropping that and just saying “is a social sparrow and is widely ….”

96: should this be “from the end of the breeding season through winter”?

99: comma splice. Avoid. (“, and”)

117, 119: What is meant by “wing length”? Wing chord? Also, how exactly is body length measured?

120: Is it necessary to say “digital photographs were taken with a digital camera”? Can you not just say “photographs were taken with a digital camera”?

Or better yet, avoid passive voice and say “We photographed …. using a digital camera….”. Of course, I’m not sure what the editorial policy of PLOS One is re passive voice.

126: Are you sure that your colour markings on the tail (or the colourbands) had no effect on agonistic outcomes? A la Nancy Burley?

129: I’m getting lots of passive voice in the Methods, so I’m assuming it’s preferred.

131: nope, here it says “we installed”

133: you mention that the aviary contains a water dish, but you don’t mention a food dish (and in the next sentence you mention that water and food were provided ad libitum.

135: vitamin-rich corn? That kind of implies that neither the millet nor mealworms were vitamin rich. I’m assuming it was cracked corn (vs. whole corn kernals). Perhaps say “vitamin-enriched cracked corn”

138: recordings were made during the breeding season? This seems unusual given that the birds probably aren’t living in flocks during the breeding season. Nor can we assume their behavior is unaffected by the fact that it is the breeding season. This is a big potential problem.

138: I don’t think it’s necessary to say that individuals were identified by their applied colour markings (since you had just finished describing the color markings in the previous paragraph). Also, the word “markings” implies that the bands provided no identification use, which I assume is incorrect.

Behavioural Analyses

141: “on” the feeding plate? Only? It seems likely that some interactions took place “beside” the plate rather than just on the plate. You ignored those?

146: you mention unambiguous displacement. But what did you do when it was not unambiguous. You should probably say that you omitted/ignored displacements that were ambiguous.

147: “comprising the total number”? Is that the word you want?

Statistical Analyses

168: use plural of “test”

Or say a Mann-Whitney test and a Kruskall-Wallis test

173: “bill-nose” length? You didn’t mention this in the Methods. I’ve never heard of “bill-nose length”. What is the “nose” of a bird anyway? What about body length (which, frankly, I’ve also never heard of).

Results

185: The fact that males engaged in more aggressive behaviors during the breeding season is not terribly surprising. Recrudesence of the gonads is occurring at this time, so presumably testosterone levels are elevated in males. I worry that keeping the birds in an aviary during the breeding season (or at least the onset of the breeding season) is more than a little problematic.

189: I’m a little confused by the “to” in “male-to-male” etc. This implies that there was always an aggressor? You implied already that only unambiguous outcomes were recorded. How did you score it if the aggressor LOST? Can you have an X to Y dyad if Y won and X lost? Even if X was the aggressor?

191: How much of this aggression could be explained by the fact that the birds in each locale might have already been familiar with one another. How long did you allow the birds in the aviary to habituate before you started collecting data?

Also, I would assume that 60 km is not enough distance for any genetic differences between populations to have an effect.

Discussion

236: comma splice (, and they established….)

I’m struck by your use of the word “established”. By definition, one should observe more threats and fewer attacks as a dominance hierarchy is established. In other words, there should be a strong temporal component involved. But I don’t recall anything about you reporting a decrease in attacks over time. I feel that this is a potential problem.

243: another comma splice

275: again, it depends on the amount of time the individuals had to get to know one another. I would suspect that after a month, flock of origin might not matter anymore.

282: unclear what is meant by “more conspicuous characteristics”.

300: fighting ability depends on fitness? Are you referring to “reproductive fitness” here? Or physical condition?

315: you are saying that individuals from the two sites were unfamiliar with one another. That may be the case during the first week, but I don’t think there’s any reason to believe that the birds did not know each other well by the end of the experiment.

326: Here you say bill length rather than bill-nose length. Also, I notice that in the Methods (line 117), you refer to the bill as the beak. Be consistent. And, as I mentioned earlier, there is certainly no mention of noses in the Methods.

Reviewer #2: This is a well-constructed paper with a straightforward study. Study design and conclusions are explained clearly. Behavioural studies are not my area of expertise, so I’m not sure if additional data should be included as supplementary information (eg. More details on the study design/data collection protocols?), but I was surprised at how little data was reported/shared. Some caveats/limitations should be discussed; eg. Does the stress of being captured and transferred to an aviary influence the number/type of aggressive behaviours? Have there been studies like this done in the wild?

Other than that, I only have minor editorial comments; a few grammatical errors should be attended to, and make sure to refer to all tables in the text (table 1 is not mentioned in the text).

- 43: “mobbing of potential predators. However…” should be new sentence

- 47-48: remove “such as”, “and through social status signalling” to “or through…”

- 52: “due to” should be “through”, remove “thus”

- 60: agonistic instead of agnostic?

- 63: can you specify what kinds of ornaments? Maybe ‘cranial ornaments’?

- 84-87: I would put these sentences together – “(status signalling trait), and individuals with large badges..”

- 88: “…are commonly occurred for occupying several resources” is an awkward grammar. Are you trying to say “conflict between groups commonly occurs when the individuals/groups occupy several resources”? Or “conflict between groups occurs for the occupation of several resources”?

- 90: often compete to protect territories from other groups, but this varies between breeding and non-breeding seasons

- 91-92: “were occurred” should be changed to “occurs”

- 93-94: this question is confusing; if it is intergroup conflict, doesn’t that mean that it is happening at the group level? If this is referring to group fitness vs individual fitness, perhaps clarify this

- 102: “not unambiguously supported” is awkward phrasing because it’s a double negative, so its unclear what this sentence means. Rephrase

- 106: remove “so far”

- 108: remove “using video footage”; presumably this is explained in the methods section; change “and we distinguished” to “do distinguish”

- 144: is it possible to describe threat displays a little more? E.g, distracting movements, flexing feathers, jumping etc.

- 159: might be good to show the equations for w and w2 as well, since the lines before the equation mention P, i, and j, but none of these are present in the equation shown

- 181: add the word ‘total’ before ‘aggressive behaviours’ to emphasize that attacks and threat displays are types of aggressive behaviours

- 187: add the numbers of aggressive encounters between male-female, male-male, and female-female dyads in brackets, or refer to the table with this data (table 1)

- 233: I would start the discussion with a bigger picture overview of the significance of this study and the results. Eg. Something about why it is interesting to study aggressive interactions in birds. Its possible you don’t need this first discussion paragraph at all, and can start with the second paragraph

- 309: signals of what during threat displays? Signals of dominance or high-status?

- 320: explain what fission-fusion groups are

6. PLOS authors have the option to publish the peer review history of their article (what does this mean?). If published, this will include your full peer review and any attached files.

Reviewer #1: No

Reviewer #2: No

---

## [Author Response · Author response to Decision Letter 0]

1 Mar 2023

Responses to the Reviewers’ comments

1. Responses to the academic editor

We highly appreciate your reviewing efforts and your detailed comments. We have modified the manuscript as per your instructions and tried to resolve all issues you mentioned. Herein, we are responding to each of your concerns point by point.

1. Regarding the study period during the early breeding season: The tree sparrow generally forms social flock in breeding season, and exhibit group foraging and group breeding in breeding season (Summers-Smith 1995, Lee et al. 2020, Lee 2022). In addition, captive sparrow populations tend to reproduce later than outdoor populations, probably due to changes in sunlight exposure and temperature. Therefore, the social behavior of the sparrows that we recorded was analyzed to before the breeding season, and the recording was stopped at the start of the breeding behavior. We have added this to the text.

Added: Line 136-137

“before the breeding season, and the recording was stopped at the start of the breeding behaviour.”

2. Regarding the familiarity and changes of agonistic behaviors by temporal axis: Thank you for your detail comments. In this study, attacking behaviors and threating behaviors were recorded the most in the first week of recording, and attacking behavior rapidly decreased, but threatening behavior decreased relatively slowly. Then, attacking behavior was slightly increased only in the first week of May, the last week of recording. Observation of the sparrow population showed that both attacking and threatening decreased as low-ranking individuals did not approach feeding areas with alpha males in the latter half of the recording period. We added this information to the text.

Added: Line 194-195

“Attack and threat displays were recorded the most in the first week, and attack displays rapidly decreased by week, but threat display decreased relatively slowly (Fig 1).”

Added: Line 211

“Fig 1. Changes of the number of attack and threat displays by week”

3. Regarding the stress influence the number/type of aggressive behaviors: Thank you for your comment. Since stress of being captured and transferred to an aviary influence the number/type of aggressive behaviours, we released individuals into the aviary after stabilizing in a dark room, and we started behavioural recording one week after the captive individuals had adapted to the aviary without human access for a week. Therefore, we judged that the stress of being captured and transferred to aviary could not influence the aggressive behaviours. We added the sentence in the manuscript.

Added: Line 133-135

“We released captured individuals into the aviary after stabilizing in a dark room, and behavioural recording was started one week after the captive individuals had adapted to the aviary without human access for a week.”

4. Providing the similar studies done under field conditions: Thank you for your comment. As we included the manuscript in the manuscript, Rat et al. (2015) was the good examples for the study of dominance hierarchy and social status signaling under the field conditions. Therefore, we emphasized the reference of these studies.

Rat, M., van Dijk, R. E., Covas, R., & Doutrelant, C. (2015). Dominance hierarchies and associated signalling in a cooperative passerine. Behavioral Ecology and Sociobiology, 69(3), 437-448. https://doi.org/10.1007/s00265-014-1856-y

Added: Line 52-53

“..in both captive and field conditions, …”

Added: Line 78-80

“The study of sociable weaver under the field condition, they exhibit an ordered hierarchy within colony and the size of the bib predicts success in social competition [9].”

5. Ensuring that manuscript meets PLOS ONE’s style requirements: Thanks for your comment. We have revised the manuscript accordingly. We have modified the symbols of the author’s affiliation, font sizes of headings and subheadings, spacing of the first lines of all paragraphs, formatted tables, etc., following the guidelines.

6. Providing the data availability statement: Thanks for your comments. We added the ‘Data availability statement’ in the manuscript.

7. Providing full ethics statement in the ‘Methods’ section: Yes, we provided full ‘Ethical Note’ in the ‘Materials and Methods’ section. 

Added: Line 183

Ethical Note

This study was approved by the Chonnam National University (CNU) and CNU Laboratory Animal Research Centre (License no: CNU IACUC-YB-2020-01) under the Association for the Study of Animal Behaviour Guidelines for the Treatment of Animals in Behavioural Research. Tree sparrows received their typical daily diets, and water was available ad libitum throughout the observations. We made efforts to minimize disturbance during behavioural data collection by recording without the presence of researchers.

2. Responses to the Reviewer 1

We highly appreciate your reviewing efforts and your detailed comments. We have modified the manuscript as per your instructions and tried to resolve all issues you mentioned. Herein, we are responding to each of your concerns point by point.

1. Line 40; Changed: Line 39

“flocks” → “groups”

2. Line 41; Changed: Line 40

“the individuals” → “individuals”

3. Line 43; Regarding the word consistency: Thank you for your suggestion. We changed all the words the same for consistency.

Word changed: forage → food resources

Changed: Line 40 

Changed: Line 56

Changed: Line 93

4. Line 53; Changed: Line 50

(loser; Drews, 1993). → (loser) [10].

5. Line 56; Changed: Line 52

“several” → “various”

Changed: Line 53

“, including…” → “Examples include…”

6. Line 56; Regarding the existence of dominance hierarchy in the social species: Thank you for your advice. According to Rat et al. (2015), dominance hierarchies are found in many taxa ranging from insect to primates, including humans, and are crucial for group stability and cohesion. However, dominance hierarchies are generally found for social animals because dominance hierarchies are marked by strong directional asymmetry between individuals forming the group that exhibit social-relationships within the group. Therefore, we remained to use this word because we judged that the dominance hierarchy occurs in social groups where interactions between entities continue (e.g. Creel et al. 2013, Functional Ecology; Rat et al. 2015, Behavioral Ecology and Sociobiology; Shizuka & McDonald 2015, Journal of the Royal Society Interface)

7. Line 61: Changed: Line 58

“,” → “;”

8. Line 65: Regarding the word “reproductive fitness”: We considered reproductive fitness as an individual’s quality which is important for reproductive success, and reproductive fitness, as you said, is included in physical fitness. Therefore, we integrated these words into physical fitness.

Changed: Line 61

“male reproductive fitness, health, fighting ability, and aggressiveness” → “physical fitness which includes male reproductive fitness, health, fighting ability, and aggressiveness”

9. Line 66: Changed: Line 61

“fitness” → “physical fitness”

10. Line 73-74: Changed: Line 68-69

“certain morphological traits” → “larger or more colourful patches”

11. Line 73-74: Rephrase sentences: Thank you for your detailed advice. We integrated and reorganized sentence.

Changed: Line 68-69

“by certain morphological traits (e.g., melanin-based plumage coloration, also referred to as ‘badge of status’; Rat et al., 2015) social status signals, and larger or more colourful patches (Senar, 2006).” → “by larger or more colourful patches [22], and melanin-based plumage coloration, (also referred to as ‘badge of status’) [9]..”

12. Line 76: Changed: Line 71

“birds exhibit” → “birds often exhibit”

13. Line 82: Changed: Line 76-77

“Eurasian siskins (Carduelis spinus) may bear a badge of dominance, and individuals with larger, blackish bib plumage are typically dominant over individuals with smaller badges.” → “Eurasian siskins (Carduelis spinus) may bear a badge of dominance with individuals with larger, blackish bib plumage are typically dominant over individuals exhibiting smaller badges.”

14. Line 88: Changed: Line 84-85

“conflicts between groups (intergroup conflict) are commonly occurred for occupying several resources” → “conflict between groups (intergroup conflict) occurs for the occupation of several resources”

15. Line 90: Changed: Line 85-87

“often compete for protecting territories against individuals form other groups, but it differs between breeding and non-breeding season” → “often compete against individuals from other groups in order to protect their territories. However, such competition can differ between breeding and non-breeding season”

16. Line 92: Changed: Line 87

“were occurred” → “were observed” 

17. Line 92: Changed: Line 88

“influence” → “influenced”

18. Line 93-94: Changed: Line 88-89

“However, it is uncertain whether intergroup competition occurs at the group level (intergroup exclusivity) or individual level (individual fitness).” → “However, the role played by group selection in such circumstances in unclear.”

19. Line 95: Providing scientific name of Eurasian tree sparrows: We already introduced scientific name of the tree sparrow (line 55), but we changed ‘The Eurasian tree sparrow’ to ‘The tree sparrow’ to unify the words. 

Changed: Line 90

“The Eurasian tree sparrow” → “The tree sparrow”

20. Line 95: Changed: Line 90

“is a typical social bird that is widely…” → “is a social sparrow and is widely”

21. Line 96: Changed: Line 91

“from the breeding season to winter” → “from the end of the breeding season through winter”

22. Line 99: Changed: Line 94

“, and” → “ and”

23. Line 117, 119: Providing measurement information: As you pointed out, wing length means wing chord length, and body length means length from the tip of the beak to the tip of the tail with the bird laid down. We measured tree sparrows after capturing individuals following morphological variables according to the Bird Banding Manual (Korea national Park Research Institute, 2007), so we followed the word in this reference and cited this reference in the manuscript (e.g. Lee et al. 2022. The Wilson Journal of Ornithology).

Changed: Line 110-111

“we measured tarsus, wing, beak, and body length, as well as badge size.” → “we measured the following morphological variables according to the Bird Banding Manual (Korea National Park Research Institute 2007).”

Changed: Line 112

“and wing and body lengths” → “and wing length (wing chord length) and body lengths”

Added: Line 113-117

“Beak lengths include bill-nose length (length from the tip of the nostril to the tip of the beak), bill-head length (length from the tip of the back of the head to the tip of the beak), and bill-skull length (length from the junction of the skull and beak to the tip of the beak). Body length is the length from the tip of the beak to the tip of the tail when the bird is laid down, and badge…”

24. Line 120: Changed: Line 117-119

“Digital photographs of each individual were taken using a digital camera (Samsung Galaxy S20+, Samsung, Suwon-si, Republic of Korea), and badge size (mm2) was measured …” → “. We photographed each individuals using a digital camera (Samsung Galaxy S20+, Samsung, Suwon-si, Republic of Korea), and measured badge size (mm2) using ImageJ software”

25. Line 126: Regarding the colour marking affects agonistic outcomes: Thank you for your detailed comments. Previously, we conducted preliminary experiment in 2019 that in which the colour of the tail and the color of the band were changed during the experiment. As a result, we found no effect on agonistic outcomes among individuals. Therefore, in this study, we used the colour markings on the tail and colour bands on the tarsus to identify individuals.

26. Line 129: Regarding to write passive voice or active voice: Thank you for your suggestion. We changed the sentences as active voice.

Changed: Line 125

“Captured tree sparrows were transferred to a semi-outdoor aviary on the rooftop…” → “We transferred captured tree sparrows to a semi-outdoor aviary on the rooftop…”

27. Line 133: Added: Line 128-129

“a food dish”

28. Line 135: Changed: Line 131

“vitamin-rich corn” → “vitamin-enriched cracked corn”

29. Line 138: Regarding the flock behaviors during the breeding season: The tree sparrow generally forms social flock in breeding season, and exhibit group foraging and group breeding in breeding season (Summers-Smith 1995, Lee et al. 2020, Lee 2022). In addition, captive sparrow populations tend to reproduce later than outdoor populations, probably due to changes in sunlight exposure and temperature. Therefore, the social behavior of the sparrows that we recorded was analyzed to before the breeding season, and the recording was stopped at the start of the breeding behavior. We have added this to the text.

Added: Line 136-137

“before the breeding season, and the recording was stopped at the start of the breeding behaviour.”

30. Line 138: Regarding the statement of “marking”: Thank you for your suggestion. We deleted the phrase.

Deleted: “and each individual was identified according to the applied colour markings.”

31. Line 141: Changed: Line 140

“We recorded the total number of aggressive behaviours of tree sparrows that occurred on the feeding plate.” → “We recorded the total number of aggressive behaviours of tree sparrows that occurred on the feeding plate and in the surrounding area”

32. Line 146: Added: Line 146-147

“We omitted and ignored ambiguous displacements.”

33. Line 147: Regarding the ‘comprising the total number’: Yes, we regarded the fighting success as the total number of aggressive interactions to calculate David’s Score. 

34. Line 168: Changed: Line 172

“we used Mann-Whitney U test and Kruskal-Wallis test.” → “we used Mann-Whitney U tests and Kruskal-Wallis tests.”

35. Line 173: Regarding the using terms: bill-nose length and body length: As we mentioned above (no. 23), we followed measuring method of morphological variables according to the Bird Banding Manual (Korea national Park Research Institute, 2007). So we added the information in the manuscript.

Added: Line 113-117

“Beak lengths include bill-nose length (length from the tip of the nostril to the tip of the beak), bill-head length (length from the tip of the back of the head to the tip of the beak), and bill-skull length (length from the junction of the skull and beak to the tip of the beak). Body length is the length from the tip of the beak to the tip of the tail when the bird is laid down, and badge”

36. Line 185: Considering male engaged in more aggressive behaviors and breeding season: Thank you for your detailed comment. Regardless of breeding and non-breeding seasons, several studies have reported that males were more aggressive than females (Cramp & Perrins 1994, Mónus et al. 2017), and of course, males showed high aggression near the breeding season, but in general, male tree sparrows are large and dominant over females. The results of male aggression in this study were similar to those of other studies, and this was consistent until right before breeding season. Therefore, in this manuscript, we added the phrase that this study was conducted from before the breeding season to onset the breeding season.

Added: in result (Line 136-137)

“before the breeding season, and the recording was stopped at the start of the breeding behaviour.

“

37. Line 189: Providing “X-to-Y” dyads: Yes, we implied that X was always an aggressor, and we recorded only unambiguous outcomes. We focused on performing aggressive behaviour. When an attack or threat (aggressive behaviour) was applied to each other, both individuals (X and Y) scored 1 because X and Y were both ‘the winner’ and ‘the loser’. We added this statement in the manuscript.

Added: Line 147-148

“If attacks occur simultaneously between each other in the X-to-Y dyad, both X and Y are considered winners and both scored 1”

38. Line 191: Regarding the aggression between individuals and habituate before recording: Thank you for your detailed comments. We gave the birds a week to acclimate, and then we started recording. Also, tree sparrows are representative resident birds in Korea and do not migrate far from their breeding grounds (Summers-Smith 1995, Chae 2019, Lee et al. 2020, Lee 2022). Therefore, the 60 km separation distance of our capture area was not to show genetic differences, but to test the interaction of different populations.

Added: Line 134-135

“and behavioural recording was started one week after the captive individuals had adapted to the aviary without human access for a week.”

39. Line 236: Regarding the establishment of dominance hierarchy: Thank you for your detail comments. In this study, attacking behaviors and threating behaviors were recorded the most in the first week of recording, and attacking behavior rapidly decreased, but threatening behavior decreased relatively slowly. Then, attacking behavior was slightly increased only in the first week of May, the last week of recording. Observation of the sparrow population showed that both attacking and threatening decreased as low-ranking individuals did not approach feeding areas with alpha males in the latter half of the recording period. We added this information to the text.

Added: Line 194-195

“Attack and threat displays were recorded the most in the first week, and attack displays rapidly decreased by week, but threat display decreased relatively slowly (Fig 1).”

Added: Line 211

“Fig 1. Changes of the number of attack and threat displays by week”

40. Line 243: Changed: Line 258

“…Mónus et al., 2016), and…” → …41, 42] and”

41. Line 275: Regarding the inter-group competition: Thank you for your detailed comment. As we described above, most of the tree sparrow's agonistic interactions occurred within early period and continued to decline. As a result of the analysis, there was no difference inter- and intra-group agonistic interactions of the tree sparrow in both the early and late periods, and there was no difference even in the early period when the agonistic interactions were concentrated. We added ‘during the research periods’ to the sentence.

Added: Line 287

“… during the research periods”

42. Line 282: Changed: Line 292-293

“Japanese quail (Coturnix coturnix) prefer novel individuals to known partners, as novel partners may show slightly more conspicuous characteristics than familiar individuals” → “Japanese quail (Coturnix coturnix) prefer conspicuous and slightly novel partners that somewhat different from familiar ones.”

43. Line 300: Changed: Line 309

“body size and fitness” → “physical fitness”

44. Line 315: Regarding the inter-group identification: Thank you for your comment. As the reviewer pointed out, we don't think each individuals of tree sparrows are not considered unfamiliar until later in the recording period. However, as mentioned above, since tree sparrows consistently showed agonistic interactions regardless of the group from the early recording period to the late recording period, we assume that individual physical fitness is more important than the group origin of tree sparrows.

45. Line 326: Regarding to use ‘beak length’: Thank you for your comments. As we mentioned above, we added a detailed explanation of beak length (Line 113-117).

Changed: Line 333

“Bill length” → “Bill-nose length”

3. Responses to the Reviewer 2

We highly appreciate your reviewing efforts and your detailed comments. We have modified the manuscript as per your instructions and tried to resolve all issues you mentioned. Herein, we are responding to each of your concerns point by point.

1. Regarding the stress influence the number/type of aggressive behaviors: Thank you for your comment. Since stress of being captured and transferred to an aviary influence the number/type of aggressive behaviours, we released individuals into the aviary after stabilizing in a dark room, and we started behavioural recording one week after the captive individuals had adapted to the aviary without human access for a week. Therefore, we judged that the stress of being captured and transferred to aviary could not influence the aggressive behaviours. We added the sentence in the manuscript.

Added: Line 133-135

“We released captured individuals into the aviary after stabilizing in a dark room, and behavioural recording was started one week after the captive individuals had adapted to the aviary without human access for a week.”

2. Providing the similar studies done under field conditions: Thank you for your comment. As we included the manuscript in the manuscript, Rat et al. (2015) was the good examples for the study of dominance hierarchy and social status signaling under the field conditions. Therefore, we emphasized the reference of these studies.

Rat, M., van Dijk, R. E., Covas, R., & Doutrelant, C. (2015). Dominance hierarchies and associated signalling in a cooperative passerine. Behavioral Ecology and Sociobiology, 69(3), 437-448. https://doi.org/10.1007/s00265-014-1856-y

Added: Line 52-53

“..in both captive and field conditions, …”

Added: Line 78-80

“The study of sociable weaver under the field condition, they exhibit an ordered hierarchy within colony and the size of the bib predicts success in social competition [9].”

3. Line 43: Changed: Line 42

“mobbing of potential predators; however…” → “mobbing of potential predators. However, ...”

4. Line 47-48: Changed: Line 45-46

“such as through establishing dominance hierarchies (Rowell, 1974) and through social status signalling (Rohwer & Ewald, 1981).” → “through establishing dominance hierarchies [7] or through social status signalling [8].”

5. Line 52: Changed: Line 49-50

“due to dyadic aggressive interactions, thus producing a dominant individual (winner)” → “through dyadic aggressive interactions, producing a dominant individual (winner)”

6. Line 60: Changed: Line 57

“agnostic” → “agonistic”

7. Line 63: Specifying the ornament: Generally, ornament of birds means the plumage coloration, ornamental feathers, and melanin-based ornament that based on the bird feather, so we changed the word ‘ornament’ to ‘plumage ornamentation’. 

Changed: Line 59-60

“ornament” → “plumage ornamentation”

8. Line 84-87: Changed: Line 81

“of dominance rank (i.e., a status signalling trait). Individuals with large badges” → “of dominance rank (i.e., a status signalling trait), individuals with large badges”

9. Line 88: Changed: Line 84-85

“conflicts between groups (intergroup conflict) are commonly occurred for occupying several resources” → “conflict between groups (intergroup conflict) occurs for the occupation of several resources”

10. Line 90: Regarding to change sentence: Thank you for your detailed advice. We changed the sentence as you and Reviewer #1’s advice.

Changed: Line 85-87

“often compete for protecting territories against individuals form other groups, but it differs between breeding and non-breeding season” → “often compete against individuals from other groups in order to protect their territories. However, such competition can differ between breeding and non-breeding season”

11. Line 91-92: Changed: Line 87

“were occurred” → “were observed”

12. Line 93-94: Clarifying the sentences: As you and the Reviewer #1 questioned, we changed the sentence as the Reviewer #1’s suggestion.

Changed: Line 88-89

“However, it is uncertain whether intergroup competition occurs at the group level (intergroup exclusivity) or individual level (individual fitness)” → “However, the role played by group selection in such circumstances in unclear.”

13. Line 102: Changed: Line 96

“not unambiguously supported” → “not clearly supported”

14. Line 106: Deleted

“so far”

15. Line 108: Changed: Line 101

“in captive tree sparrow flocks using video footage, and we distinguished” → “in captive tree sparrow and do distinguish”

16. Line 144: Added: Line 144

“(e.g. distracting movements, fluffing feathers, head-forward threat display, etc.)”

17. Line 159: Regarding to show the equations: Thank you for your suggestion. We added the details for Pij because w and l values were the simple sum of i’s Pij values (as described below). 

Added: Line 159-161

“The Pij is the number of times that i defeats j (aij) divided by the total number of its interactions between i and j (nij; Pij = aij/nij). Thus, DS for each member could calculated with this formula:”

18. Line 181: Added: Line 195-196

“The mean number of performed total aggressive behaviours”

19. Line 187: Regarding to add the number oaf aggressive encounters: Thank you for your suggestion. We refer to the table with this data (Table 1)

Added: Line 204

“…was the lowest (Table 1).”

Added: Line 205

“…to the number of aggressive behaviours (χ23 = 0.073, P = 0.99; Table 1)”

20. Line 233: Regarding to change paragraph: Thanks for your suggestion. We changed paragraph as your suggestion.

Changed: Line 249-256

“In this study, we investigated aggressive behaviour of captive tree sparrows in flocks, specifically considering the effects of sex, dominance hierarchy, and status signalling with respect to morphological traits. When foraging on feeding plates, tree sparrows frequently interacted through aggressive behaviours, including attacks and threat displays and they established a dominance hierarchy based on the outcomes of aggressive behaviours. Males showed higher dominance ranks than females, but no differences were observed regarding capture-site groups. We found that morphological characteristics independently affected social status signalling; DSattack was significantly correlated with bill¬-nose length, whereas DSthreat was significantly correlated with sex and badge size.” → “Competition is essential for birds to occupy limited food resources, and agonistic interactions are common, especially in social birds living in groups. In this situation, the social system was characterized by dominance hierarchy to mediate conflicts and the social status signal has evolved to reduce the cost by agonistic interaction [41-42]. When foraging on feeding plates, tree sparrows frequently interacted through aggressive behaviours, including attacks and threat displays and they established a dominance hierarchy based on the outcomes of aggressive behaviours. We found that morphological characteristics independently affected social status signalling, and it is likely to be important for organizing stable society to mediate conflicts in the tree sparrow flock.”

21. Line 309: Added: Line 317

“…sex and badge size are signals of high-status during…”

22. Line 320: Providing what fission-fusion groups are: We intend to use the ‘fission-fusion group’ as large group whose members were changed constantly, so we added the supporting phrase after the word. 

Changed: Line 326-327

“tree sparrows frequently form large fission-fusion groups, sometimes comprising hundreds of individuals” → “tree sparrows frequently form large groups (sometimes hundreds of individuals), whose members were changed constantly”

---

## [Editor Report · Decision Letter 1]

13 Mar 2023

Morphological characteristics convey social status signals in captive tree sparrows (*Passer montanus*)

PONE-D-22-22925R1

Dear Dr. Sung,

We’re pleased to inform you that your manuscript has been judged scientifically suitable for publication and will be formally accepted for publication once it meets all outstanding technical requirements.

Kind regards,

Ofer Ovadia

Academic Editor

PLOS ONE
---

## [Editor Report · Acceptance letter]

20 Mar 2023

PONE-D-22-22925R1 

Morphological characteristics convey social status signals in captive tree sparrows (*Passer montanus*) 

Dear Dr. Sung:

I'm pleased to inform you that your manuscript has been deemed suitable for publication in PLOS ONE. Congratulations! Your manuscript is now with our production department. 

Kind regards, 

on behalf of

Dr. Ofer Ovadia 

Academic Editor

PLOS ONE